# Table 2 Fallacy in Descriptive Epidemiology: Bringing Machine Learning to the Table

**DOI:** 10.3390/ijerph20136194

**Published:** 2023-06-21

**Authors:** Christoffer Dharma, Rui Fu, Michael Chaiton

**Affiliations:** 1Dalla Lana School of Public Health, University of Toronto, Toronton, ON M5T 3M7, Canada; chris.dharma@mail.utoronto.ca (C.D.); rui.fu@mail.utoronto.ca (R.F.); 2Center for Addictions and Mental Health, Toronto, ON M6J 1H4, Canada; 3Department of Otolaryngology—Head and Neck Surgery, Temerty Faculty of Medicine, Sunnybrook Hospital, Toronto, ON M4N 3M5, Canada; 4Ontario Tobacco Research Unit, Toronto, ON M5S 2S1, Canada

**Keywords:** machine learning, descriptive analysis, data analysis methods, alcohol use, sexual minority youth

## Abstract

There is a lack of rigorous methodological development for descriptive epidemiology, where the goal is to describe and identify the most important associations with an outcome given a large set of potential predictors. This has often led to the Table 2 fallacy, where one presents the coefficient estimates for all covariates from a single multivariable regression model, which are often uninterpretable in a descriptive analysis. We argue that machine learning (ML) is a potential solution to this problem. We illustrate the power of ML with an example analysis identifying the most important predictors of alcohol abuse among sexual minority youth. The framework we propose for this analysis is as follows: (1) Identify a few ML methods for the analysis, (2) optimize the parameters using the whole data with a nested cross-validation approach, (3) rank the variables using variable importance scores, (4) present partial dependence plots (PDP) to illustrate the association between the important variables and the outcome, (5) and identify the strength of the interaction terms using the PDPs. We discuss the potential strengths and weaknesses of using ML methods for descriptive analysis and future directions for research. R codes to reproduce these analyses are provided, which we invite other researchers to use.

## 1. Introduction

### 1.1. Descriptive Epidemiology

Fox et al. in the American Journal of Epidemiology called for the need to revitalize descriptive epidemiology [1], which refers to studies that are “more concerned with describing associations than with analyzing and explaining causal effects” [2]. Descriptive epidemiology, since John Snow developed his maps of cholera, is a core and vital component of epidemiology. Overwhelmingly, the most important advances in our understanding of public health and the factors affecting public health have come first from descriptive epidemiology. It is the discovery of patterns in person, place, and time that have driven our critical public health achievements from curtailing infectious diseases through clean water, clean air, and sanitation as well as the identification of chronic disease risk factors. Causal epidemiology, by contrast, which is concerned with quantifying the effect of an exposure or treatment on the outcome, is a secondary step that refines and tests what we have already discovered from our descriptive epidemiology [2].

Unfortunately, descriptive epidemiology has suffered from perceptions that it is less prestigious than other forms of epidemiology. Consequently, there exists significant and problematic literature that attempt to “go beyond” descriptive analysis by performing the types of analysis that are meant for causal analysis. We have all observed these studies with the same general framework: conducting a descriptive analysis, presenting bivariate associations, and then entering all or a subset of the variables into a logistic regression. The results of the regression are then presented in a Table 2, and each variable is interpreted. This interpretation of the estimates of the covariates in the model rather than the estimates of the focal variable is known as the Table 2 fallacy [3].

There is a reason that the Table 2 fallacy is so widespread. Descriptive epidemiology is attempting to describe the associations between multiple variables and identify patterns within the data. This is often known as “risk factor” epidemiology [1]. The goal of these risk factor analyses is often to find the most important predictors or risk factors associated with an outcome among a large number of predictors. Entering all the variables into a logistic regression appears to solve this issue—the variables that continue to be statistically significant in these kitchen-sink regressions are said to be “independently associated” with the outcome and, consequently, important. These reasons for conducting this type of regression are valid—it would be helpful to have methodologies that comprehensively assess the patterns in the data, and it is necessary to have a methodology that identifies the most important variables where there are no a priori hypotheses.

Fundamentally, however, these kitchen-sink regressions are flawed. First, presenting results in a Table 2 as described above can be misinterpreted as answering a causal question, where one might erroneously interpret the effect of each covariate as a causal total unbiased effect. A single multivariable regression model cannot accommodate the need to adjust for all the different potential pathways for each of these covariates. This is because one variable that needs to be adjusted for as a covariate in one pathway might be a collider or a mediator in another pathway, which will create a biased estimate when it is adjusted for [3]. One might suggest to just present the unadjusted estimates for all variables in a descriptive study, but this does not resolve the biases that are present in these associations. In fact, if the interest is to determine the unadjusted association for each of these variables, one might just perform chi-square or t-tests with a set of adjusted p-values to examine the raw association of each covariate and outcome separately. Second, beyond the issues involved in the interpretation of the effect estimates of multiply adjusted covariates, the adjustments themselves can be inappropriate as it creates a counterfactual world in which all else is equal, when for descriptive epidemiology, we want first to describe the world as it is [1]. When the purpose is to rank the importance of predictors, it can be inappropriate to make these adjustments [4].

What is needed then is a new standard methodology for descriptive analysis that allows for the identification of “important” variables. New frameworks have been proposed for descriptive epidemiology [4]. Stratification or machine learning has been proposed as a potential solution for descriptive epidemiology [4,5]. In this paper, we focus on the use of machine learning methodology to explore the effect of multiple variables without implying causality. This methodology offers a more sophisticated and comprehensive analysis; it provides a method of identifying the important variables.

### 1.2. Machine Learning

Machine learning (ML) is a class of methods that is derived from computer science and artificial intelligence, where a model “learns” from the data to improve its performance. The distinguishing feature from traditional statistical inference models is that ML methods typically focus on predictive accuracy rather than making statistical inferences about a hypothesis. This is known as supervised machine learning where data are labelled, and there exists a singular outcome that the algorithm is trying to predict. Examples of methods in ML include decision trees, random forest, neural network, least absolute shrinkage and selection operator (LASSO), and elastic net regression.

Bi et al. identified a number of roles for ML in epidemiology: causal inference, diagnostic and predictive models, genome-wide studies, geospatial applications, text mining, and forecasting of infectious disease [6]. We argue that one of the best (and perhaps most intuitive) applications for ML is within descriptive epidemiology. We argue that ML models should be a standard component of many or most descriptive epidemiology studies as a replacement for the Table 2 fallacy-plagued regression models. Indeed, ML has been increasingly applied in health research, [7,8,9,10] although a systematic justification of using these methods for descriptive epidemiology purposes and an analytical pipeline to do so are currently lacking. Specifically, substance use appears to be an area with lots of interest in establishing ML as a routine statistical method; this may be attributed to the increasingly complex public health environments and large volume of data that have become available [11,12].

ML provides a few advantages over traditional regression models when running descriptive studies with the purpose of identifying the most important predictors of an outcome. First, running traditional regression can be challenging when there are a large number of variables that, according to a theoretical framework, constitute as potential predictors; the regression model may not even converge when there are more predictors than the sample size. ML methods can model the data even when there are more predictors (*p*) than the sample size (*n*), known as *p* >> *n* cases [13]. Given the ability to handle *p* >> *n* cases, ML methods are also equipped to identify complex and a large number of interaction terms that is not possible in traditional regression models when the sample size is small. Second, there are no distributional assumptions required in ML; traditional regression models have a distributional assumption of the population parameters, which may not always be met. Finally, ML allows the ranking of variables to determine their importance. The idea of variable importance (VI) is based on model reliance; if the values of all observations for a variable were shuffled randomly, how much does the model error change compared to when the data were kept as they were? If the variable is not important, then the error would not change by a large amount even if we shuffle the order of observations [14,15]. In tree-based methods, the relative importance of predictors can also be considered as the sum of the squared improvements over all internal nodes of the tree [16]. For random forest (RF), which is a tree-based ensemble method, there is a natural VI measure based on the leftover out of bag (OOB) data [14,15]. In fact, a model-agnostic measure of variable importance can be extended to all modelling techniques, including traditional regression models [14,15,17].

While it is also possible to calculate variable importance scores using non-ML methods, VI scores are a natural extension of some ML models and are much more straightforward to implement post hoc. In logistic regression, the commonly obtained effect measures are odds ratios (OR), which do not tell the relative importance of a predictor, as these will all be in different scales. Instead of presenting ORs from each variable that are not interpretable, ML methods provide actual scores of relative importance between each variable, exactly the goal of a descriptive analysis. The following example will illustrate the benefit of using ML over logistic regression.

## 2. Illustrative Example

### 2.1. Data Description

We present an example of how ML can provide more interpretable results than the Table 2 fallacy-plagued regression models through real-life data that identity risk factors for frequent alcohol abuse among LGBTQ+ youth on 70 potential predictors. A brief literature review showed an abundance of recent studies on this similar topic (including some from our team) that performed regression models within the context of both causal and descriptive analyses while presenting effect estimates in a Table 2 format as described above [18,19,20,21,22]. While there were a few past studies that used ML analysis to identify the most important predictors of alcohol use among youths [23,24,25], fewer studies have performed it specifically on LGBTQ+ youths, which might have unique factors that can lead to substance use (e.g., outness or internalized homophobia). Data were obtained from the ongoing longitudinal study LGBTQ+ Tobacco Project Survey, a survey on smoking cessation among LGBTQ+ young adults aged 16–29 years living in the provinces of Ontario and Quebec, Canada. Baseline data were collected from November 2020 to January 2021 during the coronavirus disease 2019 (COVID-19) pandemic. Previous analysis of this data and recruitment information can be found elsewhere [26,27]. There were 1414 participants used in this analysis. Ethics approval was obtained from the Centre for Addiction and Mental Health Research Ethics Board. Written consents were obtained prior to survey completion. No identifying information from the participants were collected. Missing data were imputed using the multiple imputation chained equation method (MICE) with the predictive mean matching (PMM) method [28]. The list of variables used and more information can be found in the supporting information (Appendix A) and the Github page.

### 2.2. Machine Learning Analysis

We illustrated our analysis using four common machine-learning classification methods: random forest (RF), neural network, LASSO, and elastic net. We also ran a traditional logistic regression analysis to illustrate the advantages and disadvantages of these methods. More information on these ML methods can be found in previous publications [6,29,30,31].

For the descriptive analyses with ML to ensure that we are using all of the data, we advocate for a nested cross-validation (NCV) approach; hyperparameter tuning was iterated with an outer and inner cross-validation so that the model was optimized using all the data rather than only 70% of the data. The NCVs were run with five inner CV and ten outer CV with a random hyperparameter optimization [32,33]. Model performances were evaluated using the area under the curve (AUC), which was optimized in each of the inner and outer validation steps. AUC is ideal to be used when data are balanced (i.e., outcome is not rare) [34], which is appropriate in the current example. The parameters that provided the best performance in the outer validation were chosen, and the final model was fit with all the data using the chosen optimized parameters. More information on nested cross-validation can be found in other sources [32,35].

The top 10 most important variables from the final NCV models were presented as the most important predictors of the alcohol abuse in this dataset, each with an associated variable importance (VI) score. We used a model-agnostic approach to calculate the VI scores, which are not model-dependent and allow us to compare results across multiple ML models; interested readers are referred elsewhere for alternative model-specific methods to compute the VI score [15,17]. To characterize the uncertainties associated with these VI scores, we recommend presenting them with a confidence interval (CI). There is no empirical closed form to calculate confidence intervals for variable importance to our knowledge. We recommend using a form of bootstrapping to calculate the intervals. Here, we took 100 bootstrap samples (with replacement) from the data; all samples were fitted using the optimized NCV parameters, resulting in different VI scores for each bootstrap sample [36]. The CI for each of the important variables determined previously was calculated by finding the corresponding percentiles from the bootstrap distribution [37]. The marginal effects of the variables that were identified to be important by each model were also illustrated using partial dependence plots (PDP), which computed the average predicted values at different possible values for each variable [38]. PDPs provide a visual representation on the direction and magnitude of the association for each identified variable [38].

Finally, we explored the potential two-way interactions between the variables. We focused on the interactions between the demographic variables (i.e., age, sexual orientation, gender, education, and income) as well as the interactions between these demographic variables and the top 10 most important predictors identified by each respective ML model [26,27]. Identification of the interactions was performed on the final optimized model through the NCV; the strength of the interactions was determined using Greenwell’s PDP method [14,38]. For the LASSO and elastic net, we ran all two-way interactions, which can be easily performed using the glmnet package. Some of these interactions were depicted using PDPs for interpretability [38]. We recommend readers to refer to multiple existing resources for examining potential interactions in descriptive analysis [39,40,41].

All statistical analyses were performed using R (v4.2.1, R Core Team, Vienna, Austria). The LASSO and elastic-net regression were fitted using the glmnet package, while the neural network and random forest models were fitted using the caret package [42,43]; finally, the variable importance scores were calculated using the VIP package [14]. The full code can be found on the project’s Github page. A step-by-step guide for how to conduct ML analysis for descriptive studies can be found in Table 1.

## 3. Results

The final ML models tuned from NCV had good performances with an AUC ranging from 0.75 to 0.85 (see Table 2). The running time of the models ranged from 5 min to approximately 10 h with the longest running model being the bootstrap for the random forest, while the shortest one was the LASSO model. The model with the best AUC performance was the neural network followed by the elastic net; the sensitivity and specificities between the models varied.

The variable importance scores for all top ten predictors identified by each algorithm with its respective bootstrapped confidence intervals (CI) are shown in Figure 1. The CIs were the most consistent for the random forest model. There were more variations in the CIs for the LASSO and elastic net models; the lower bounds for some variables reached 0, and the upper bounds reached 100%. The random forest model ranked overall outness, age, and internalized homophobia scores highly. The neural network model ranked cannabis use, cocaine use, and suicidality scores highly. Adverse childhood experience (ACE) was the only variable selected in the top ten by both the random forest and neural network models. The most important predictors selected by the LASSO and elastic net models were generally very similar; this includes variables such as cannabis use, cocaine use, and sexual orientation. Cannabis and cocaine use were the variables that were selected by the neural network, LASSO, and elastic net. For comparison purposes, we also illustrated the variables that would have been considered important in a traditional logistic regression model (see Appendix A). With 63 predictors and 1414 observations, the model still converged; however, some of the confidence intervals cannot be calculated with the traditional means. Nevertheless, there were similarities in the variables that emerged to have the strongest coefficients with alcohol abuse, such as GHB use, cannabis use, cocaine use, and sexual orientation.

The relationship between each selected predictor with the outcome alcohol abuse can be viewed in the PDPs shown in Figure 2a–c. The PDPs showed that, for some of the continuous measures, the relationship was curvilinear. For example, it appears that those with the highest risk for alcohol abuse were those at the highest and lowest levels of ACE; those who had average levels of ACE were the ones at the lowest risk for alcohol abuse. However, the neural network result showed a more linear pattern between ACE and alcohol, where higher levels of ACE increased the risk for alcohol abuse. The same pattern was observed for suicidal thinking. The elastic net model highlights the strongest associations with the categorical variables. Some notable ones include frequent cannabis use, gay sexual orientation, cocaine use, GHB use, and those who had quit smoking were more likely to have had alcohol abuse. These patterns were replicated by the neural network results for the variables that were also selected by both methods (e.g., cocaine use and cannabis use).

**Figure 1 ijerph-20-06194-f001:**
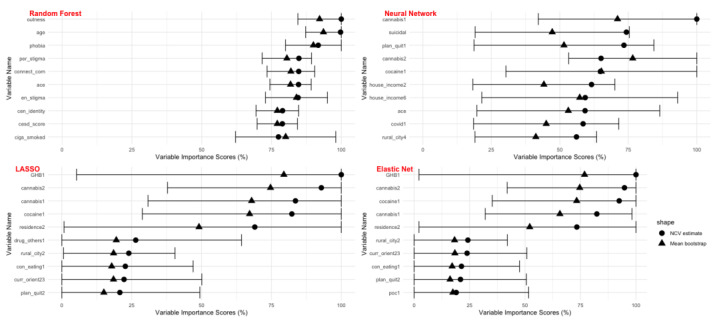
Variable importance scores from the four machine-learning models with bootstrapped confidence intervals.

The results of the interaction scores varied significantly between the models and were often inconsistent (see Appendix A for details). The random forest model was the one least likely to detect interactions; all two-way interactions investigated were very weak, as the strengths were all less than 0.01. The neural network model detected slightly stronger interactions; the strongest one found was for cocaine use and rurality, where the interaction strength was 0.11. Finally, the elastic net model detected the strongest interactions, where cannabis and employment were found to be the strongest interaction with 0.20. We illustrated two of the strongest interactions from different types of variables that were identified by the neural network model using partial dependence plots. Figure 3a shows the interaction between two categorical variables; the risk for alcohol abuse between those who have used cocaine and did not use cocaine differed by the type of cities they lived in, although the direction remained consistent, and those who used cocaine were more likely to abuse alcohol regardless of the type of city they resided in. Figure 3b shows an interaction between a continuous and a categorical variable; here, there was a heterogeneity of effect. Increases in suicidality among non-cannabis users increased the likelihood of alcohol abuse; however, increases in suicidality among cannabis users decreased the likelihood of alcohol abuse. Other interactions of interest can be depicted in a similar way with these PDPs. No meaningful interactions between continuous variables were detected in this analysis, but these can also be depicted in contour plots or three-dimensional plots [44].

## 4. Discussion

Our analysis illustrates that ML can be an invaluable tool for descriptive epidemiology, where the most important variables can be identified through a ranking process rather than determined through the size of coefficients from an incorrectly specified multivariable logistic regression. The ML method also allows for the automatic selection of important predictors from a large set of potential variables as well as easy identification of important interactions between the variables, which is more difficult for regression models to do. Moreover, this approach avoids the Table 2 fallacy where the odds ratios for every potential predictor are presented and become uninterpretable.

There are a few drawbacks that we might face with this approach. First, the question of interpretability now lies on which ML algorithm to rely on, as they all produced a slightly different set of results. Our results showed that the random forest tended to select the continuous predictors, while the LASSO tended to select categorical predictors. The LASSO is also known to be the least stable [45] and often produced results that were not always the same for different iterations, as illustrated by their least stable confidence intervals. Different instances of the single split sample from the LASSO may select very different variables; NCV may reduce this possibility, albeit the intervals were still wide as observed. Other more complex adaptations of the LASSO can also be considered to improve its stability [45,46] or the use of the elastic net as shown in this analysis. Although previous studies show that a single split sample provides results similar to those obtained from an NCV in terms of predictive performance [47], we continue to recommend the use of NCV to increase confidence in the results, particularly for descriptive analysis. This is preferred to the single split sample (usually with a ratio of 70–30) ML models, where only 70% of the data were used in model building and were not representative of the overall data [48]. With NCV, the final model is obtained using the parameters optimized using NCV and trained on all of the data to maximize the efficiency in sample size.

It is unclear if the wide bootstrapped CIs observed were the result of the relatively small sample size used in this analysis. Notably, the coefficients obtained from the traditional logistic regression model (see results in Appendix A) also resulted in a wide CI, suggesting that this is unlikely to be an ML-specific issue and might be due to the sample size. These ML analyses should be replicated with larger sample sizes to determine if resulting bootstrapped CIs from variable importance is more consistent when larger datasets are used. As the purpose of descriptive epidemiology is often exploratory, wide CIs are expected, which may also be the result of unknown complex interactions between multiple variables. Our example analysis identified some unique interactions, which may not be previously known (e.g., suicidality x cannabis use on alcohol abuse among LGBTQ+ youths), and future studies may want to further explore these interactions with larger sample sizes to confirm or disprove these associations and better understand the mechanisms.

Although the use of multiple methods poses some ambiguity in making a definite conclusion for determining the most important variables, we continue to recommend using multiple ML algorithms whenever feasible to assess the robustness of the results. The purpose of a descriptive study is also often exploratory where potential predictors are put agnostically with no particular hypothesis; hence, we refrain from making hard and fast rules in determining the final most important predictors. Regardless, we believe that the three things that should be considered when deciding the importance of predictors are (1) overlaps between different ML methods, important variables would ideally be selected with multiple algorithms (in this example, this includes cocaine use, cannabis use, rurality, ACE, and plans to quit smoking); (2) plausibility based on theory, past studies, and literature (keeping in mind that, often times, exploratory analyses were run due to the lack of studies in the topic); and (3) best predictive performance (in the current analysis will be the results from the neural network method, as shown in Table 2).

Another drawback is the computational intensity of these ML analyses compared to the traditional logistic regression models, especially with running multiple ML models, the NCV, and the bootstrap. There are packages such as nestedcv that perform NCV directly, but it is not optimized to run the VIP package that can be used to calculate variable importance to our knowledge [49]. The authors are currently planning to write a package that can automatically run all the analyses required for a descriptive epidemiology study as shown in this paper. When computing power is an issue, we recommend running the LASSO or elastic net; these models are the ones that require the fewest parameter tunings, have the simplest algorithms, and are the fastest to run even when bootstraps are required. Alternatively, one can make the parameter tuning faster for the random forest and other algorithms by only tuning for one parameter instead of two (e.g., only mtry, the number of variables to randomly select for each split for the random forest). Another method to be considered to reduce computational time is to use a Bayesian hyperparameter optimization, which allows the model to learn from a previous search and does not require the model to perform a manual grid search of the optimal parameter [33].

With the computational-intensive, data-hungry nature of ML models, we also acknowledge the limitation that model performance highly depends on the size and quality of the data. This may not be possible in settings with lower resources where the data infrastructure is not yet fully developed and computing power can be an issue. Regardless, we still advocate for using ML to generate variable importance scores in these settings with low-intensity computational methods such as LASSO, as it will still improve upon the traditional ways of presenting the kitchen-sink regression results. We refer interested readers to the existing literature on building responsible and ethical ML models [50].

We again recommend readers to other papers that go into deeper dives on each machine-learning algorithm; it is imperative to understand the best practice for each algorithm before running them for analysis. The complexity of ML might be one reason why it has not replaced traditional logistic regression yet; there are unique parameters to be optimized for each ML analysis, and readers unfamiliar with the methods might find this challenging. For starters, we again recommend the LASSO or elastic net, as these are the simplest machine-learning models.

A final criticism often faced by ML is its over reliance on black-box algorithms. This is a common criticism in any automated approach, including the traditional backwards, forwards, or stepwise regression approach for model selection. While automation can have its drawbacks, in an exploratory descriptive study that is meant to be hypothesis-generating, such a process might be desired. This can make the process of scientific exploration faster, resulting in quicker hypothesis generation for other more rigorous, in-depth causal inference investigations. The same codes can be easily recycled, which is the same when running a logistic regression for multiple analyses, except we are less worried about multiple hypothesis testing or incorrectly making causal inferences. We do not advocate for the over reliance on ML methods for causal inference.

## 5. Conclusions

Our example has illustrated how ML methods can be used to mitigate Table 2 fallacy in descriptive epidemiology through variable importance scores and automatic variable selection. We have provided reproducible codes on the Github page that can be repurposed for other studies and are currently developing a package that will be easily accessible by researchers. Researchers need to be mindful of the limitations of these methods relating to interpretability, computational intensity, and the nuances of building and validating ML algorithms, especially in low-resource settings where the volume and quality of data are less than ideal. Future research may want to further examine the ideal method to determine the most important variables in descriptive studies that will produce consistent results and make them more efficient. This is the first step to address how ML can be used in descriptive epidemiology to reduce the use of odds ratios and Table 2 fallacies in descriptive studies and instead focus on variable importance scores.

## Figures and Tables

**Figure 2 ijerph-20-06194-f002:**
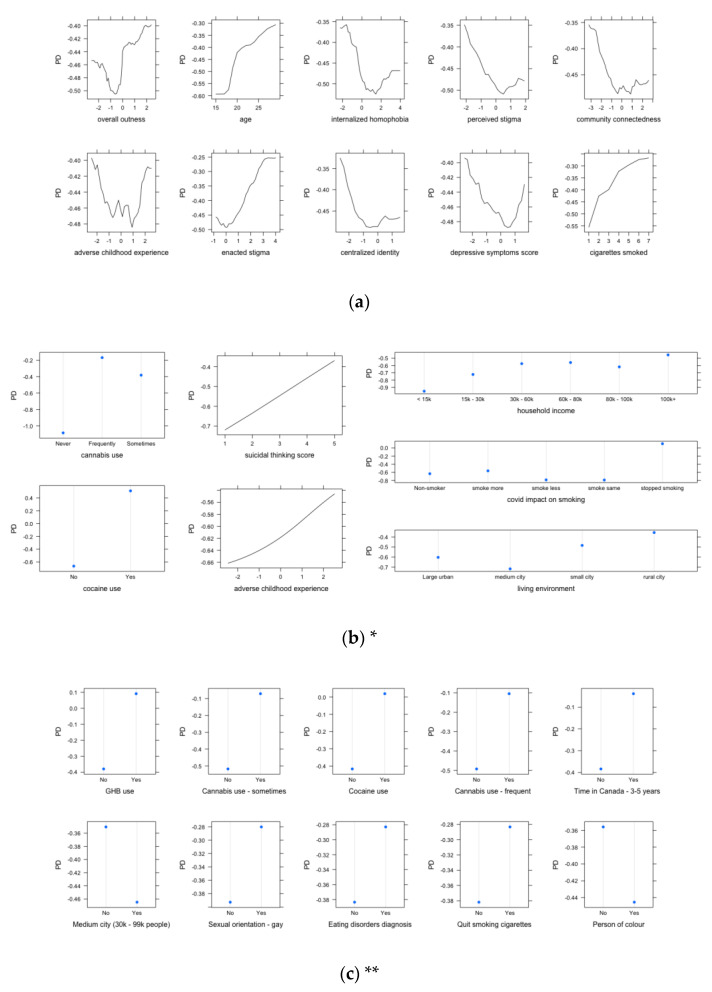
(**a**) Partial dependence plot for the random forest model. (**b**) Partial dependence plot for the neural network model. * Note: As can be observed in Figure 1, some of the selected categorical variables were dummy coded; hence, the total number of unique variables shown does not add up to 10 (e.g., cannabis1—“frequent use” and cannabis2—“sometimes use” were both selected by the neural network and elastic net models but counted as two separate variables in the top 10 most important predictors). Hence, in Figure 2b, there were fewer than 10 variables that were displayed since some were just different categories within the same categorical variable. (**c**) Partial dependence plot for the elastic net model. ** Note: Similar to * in the neural network model, the variables selected in the top 10 variables for the elastic net model were also dummy coded. However, the LASSO required a dummy-coded model matrix as the input, and hence, we displayed all the categories from the same categorical variable separately and only showed the comparison for the dummy variable for that particular category that was selected by the elastic net (e.g., rather than showing all sexual orientation categories, we only showed “gay”, which was the one selected by the model). As the results for the LASSO and elastic net were very similar (see Figure 1), we only displayed the PDPs for elastic net here. Elastic net is also a generalization of the LASSO algorithm, so showing all results is redundant.

**Figure 3 ijerph-20-06194-f003:**
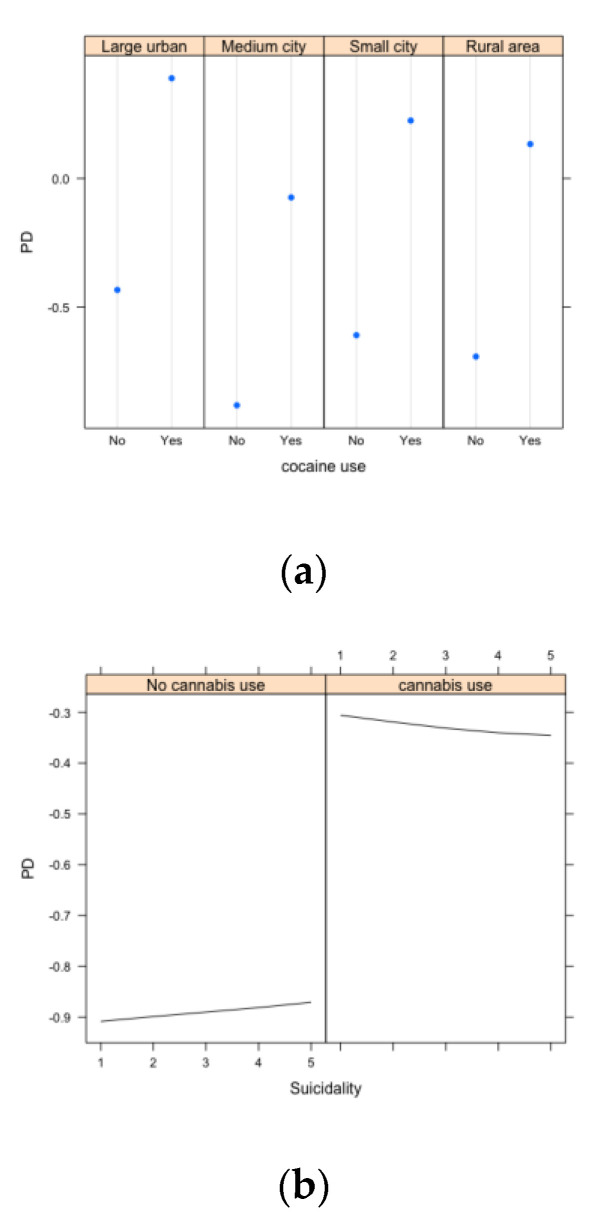
Example of partial dependence plots depicting the significant interactions detected from the neural network model. (**a**) Interactions between two categorical variables (cocaine use x rurality on alcohol abuse). (**b**) Interactions between a categorical variable and a continuous variable (cannabis use x suicidality score on alcohol abuse).

**Table 1 ijerph-20-06194-t001:** Proposed framework for conducting a descriptive analysis with machine learning.

1. Select and determine the machine-learning algorithms to be used (ideally, use at least 3; if computing time is a concern, use elastic net or LASSO).
2. Run the analysis with these machine-learning algorithms and optimize the parameters using nested cross-validation (NCV) with a predefined performance metric (e.g., AUC, accuracy, MSE). Record the performance metric from each outer validation. Ideally, use at least 5 inner validations and 10 outer validations. A random grid search can be used to find the parameters, or more advanced methods such as the Bayesian grid search can be used to save time.
3. The parameters resulting in the best predictive performance in the outer validation will be the optimized NCV parameters. The performance metric obtained from this best outer validation test will be considered as the performance of the particular ML model.
4. Fit the optimized parameters obtained from the NCV on all of the data.
5. Calculate the variable importance (VI) scores using the Greenwell method using the final model obtained in Step 4.
6. Calculate the bootstrapped confidence interval of the variable importance scores by taking a bootstrap sample of the data repeatedly and fit the NCV-optimized parameters using the ML models on the bootstrapped samples (at least 100 bootstrap replicates).
7. Plot the bootstrapped confidence interval, showing both the VI obtained in step 5 and the mean VI scores from the bootstrap (see Figure 1 as examples).
8. Show the partial dependence plots (PDP) for each of the variables that were deemed important by the ML models. PDPs will show the actual relationship between the variables and outcomes.
9. If of interest, calculate the strength of interaction using the Greenwell method also on the optimized NCV model from Step 4.
10. Plot some of the strongest interactions of interest with the PDP.
11. Consider the variables that were selected by multiple ML methods and the variables selected by the ML method with the best predictive performance to be the variables that are most importantly associated with the outcome.

**Table 2 ijerph-20-06194-t002:** Model performances from the best outer validation runs of the four machine-learning models.

	Random Forest	Neural Network	LASSO	Elastic Net (EN)
Sensitivity	0.94 (0.88, 0.98)	0.93 (0.91, 0.95)	0.94 (0.87, 0.98)	0.91 (0.89, 0.93)
Specificity	0.40 (0.27, 0.55)	0.72 (0.68, 0.76)	0.45 (0.32, 0.59)	0.40 (0.36, 0.45)
AUC (95% CI)	0.74 (0.68, 0.86)	0.85 (0.83, 0.88)	0.78 (0.70, 0.86)	0.78 (0.71, 0.86)

## Data Availability

The de-identified, anonymized data used for the analysis can be found on this link: https://borealisdata.ca/dataset.xhtml?persistentId=doi:10.5683/SP3/OPQZWQ, accessed on 15 May 2023). Codes to reproduce analysis can be found on the project’s GitHub page (https://github.com/chrisdharma/ML-descriptive-epi, accessed on 15 May 2023).

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
