# Peer review of "Table 2 Fallacy in Descriptive Epidemiology: Bringing Machine Learning to the Table"

_ijerph, 2023, doi:10.3390/ijerph20136194_

Round 1

Reviewer 1 Report

CENTRAL AND GENERAL ISSUES

Summary

In this paper the authors indicate that there is a lack of rigorous methodological development for descriptive epidemiology. In this way, the authors indicate that there are arguments to affirm that machine learning (ML) is a potential solution to this problem. The paper shows the power of ML by taking alcohol abuse among sexual minority youth as an example. Finally, the potential strengths and weaknesses of using ML methods for descriptive analysis and future directions for research are discussed. These findings have implications for descriptive epidemiology. I believe that the result adds value to the literature in this field of study. However, I think there are some important aspects that need to be improved before recommending its publication in International Journal of Environmental Research and Public Health.

Specific Comments

1. The authors indicate that machine learning (ML) is a solution for descriptive epidemiology. This is a trend in other fields that are also related to epidemics such as drugs and smoking. In this sense, two recent articles point out that "environments are increasingly complex and require advanced analysis techniques, such as Machine Learning or Artificial Intelligence". The authors must include a paragraph in the introduction that indicates that there is already previous literature on smoking and drug use.

2. The authors indicate that with ML they seek to describe and identify the most important associations with an outcome given a large set of potential predictors. There is a recent article that does this for the tobacco epidemic. This paper analyzes which predictors are the most important to explain tobacco consumption, applying Machine Learning techniques. Authors should cite the article.

3. In the conclusions I do not see any paragraph showing the limitations of this work. It would be important for the limitations of this paper to be made clear.

Author Response

Thank you for considering our paper " Table 2 Fallacy in Descriptive Epidemiology: Bringing Machine Learning to the Table" for publication into International Journal of Environmental Research and Public Health (IJERPH).  We appreciate the feedback provided by the reviewers. A revised manuscript is attached that addresses and carefully incorporates each of these comments. Below are our responses to the comments and feedback from the reviewers.

Summary

In this paper the authors indicate that there is a lack of rigorous methodological development for descriptive epidemiology. In this way, the authors indicate that there are arguments to affirm that machine learning (ML) is a potential solution to this problem. The paper shows the power of ML by taking alcohol abuse among sexual minority youth as an example. Finally, the potential strengths and weaknesses of using ML methods for descriptive analysis and future directions for research are discussed. These findings have implications for descriptive epidemiology. I believe that the result adds value to the literature in this field of study. However, I think there are some important aspects that need to be improved before recommending its publication in International Journal of Environmental Research and Public Health.

Thank you for this comment

Specific Comments

1. The authors indicate that machine learning (ML) is a solution for descriptive epidemiology. This is a trend in other fields that are also related to epidemics such as drugs and smoking. In this sense, two recent articles point out that "environments are increasingly complex and require advanced analysis techniques, such as Machine Learning or Artificial Intelligence". The authors must include a paragraph in the introduction that indicates that there is already previous literature on smoking and drug use.

Thank you for this comment. We agree that there has been literature that uses machine learning on smoking and drug use, but the findings are not the focus of the paper. We are not stating that our method has not been applied in the particular question of drugs and smoking, but we are using it as an example to show how ML can be used in descriptive epidemiology, noting that previous studies may have used this method in the field. Descriptive epidemiology is not a specific field subject matter unlike drugs and smoking, it is an umbrella term that as Fox described as, studies that are “more concerned with describing associations than with analyzing and explaining causal effects”. This can include any types of health problems, including drugs and smoking. The contribution of this paper is that, many epidemiologic studies whose aims are only to describe associations often showed results from a saturated logistic regression model, where the associations are uninterpretable. A machine learning model with variable importance can help resolve this issue. This is inherent in many fields, including alcohol and tobacco, which we also gave a few examples from other past studies on substance use among sexual minorities (see references 11-15). This is the issue we are trying to resolve not field specific to substance use or smoking, we hope this clarifies the issue or any confusion that may arise. Regardless, we have added a few sentences on how ML has been used in many health research fields such as substance use in Section 1.2 page 3.

  1. The authors indicate that with ML they seek to describe and identify the most important associations with an outcome given a large set of potential predictors. There is a recent article that does this for the tobacco epidemic. This paper analyzes which predictors are the most important to explain tobacco consumption, applying Machine Learning techniques. Authors should cite the article.

Thank you for pointing us to this important paper. We have enhanced section 1.2 Machine Learning at the top of page 3 to give a more fulsome overview of current applications of machine learning in descriptive epidemiology. These empirical studies form the foundation of our methodology paper where we aim to give a systematic justification on using ML for descriptive epidemiology as well as a complete analytical pipeline to do so. Regardless, we believe there may not be an article yet on the topic of alcohol use with sexual minority youth that uses machine learning. Indeed, our team has published a scoping review on applications of ML in mental health and substance use among the LGBTQ2S+ population and did not identify any existing paper on alcohol use; we have included this review paper in this section as reference #10.

Kundu A, Chaiton M, Billington R, Grace D, Fu R, Logie C, et al. Machine Learning Applications in Mental Health and Substance Use Research Among the LGBTQ2S+ Population: Scoping Review. JMIR Med Inform. 2021 Nov 11;9(11):e28962.

  1. In the conclusions I do not see any paragraph showing the limitations of this work. It would be important for the limitations of this paper to be made clear.

Thank you, the limitations have been discussed throughout in the Discussion section. The conclusion section tends to be short with one paragraph and does not have a paragraph specifically on limitations. In the conclusion, we have discussed how future research should find other ML methods that can produce more consistent results and maybe less considered as a black box, which is a major part of the limitations. On page 9 of the conclusion section, we added a 1-sentence summary of specific limitations of using ML in descriptive epidemiology we have discussed previously.

Reviewer 2 Report

The authors present a method using Machine Learning to analyze descriptive epidemiologic data in a way that avoids the Table 2 fallacy, where all individual coefficients are erroneously interpreted as isolated, unconfounded causal effects without considering their context for inclusion in the model. Table 2 fallacy is unfortunately common in many contexts of data analysis, so I am glad to see research aiming to address this issue. I do think further clarity is needed regarding why it is the Machine Learning process that avoids Table 2 fallacy rather than the use of variable importance measures.

While I agree that naively running a “kitchen sink” regression model and treating every model coefficient as an interpretable isolated effect is not a best practice, more information and rationale is needed to convince the reader that Machine Learning is needed to avoid Table 2 fallacy as opposed to other methods of variable selection outside of Machine Learning.

MAJOR COMMENTS

1. In order for the proposed Machine Learning process to be viewed as a viable alternative to typical multivariable regression models that is not subject to Table 2 fallacy, more explanation is needed regarding why the ML process is crucial. Other methods for assessing relative importance of predictors exist outside of machine learning. Currently, this paper provides good justification for why analyses beyond a multivariable regression are needed to compare relative impact of different predictors in order to identify ones that warrant further investigation. However, it is not clear why this ML method is needed and why variable importance cannot be assessed without it.

2. Section 1.2 mentioned multiple potential methods for assessing variable importance. The methods section does not specify what one was used or why it is a valid measure of influence of a predictor on the outcome of interest. Otherwise, this method is presented as a “black box” that is not subject to Table 2 fallacy because the authors claim that is the case.

3. In the results (lines 211-224 and Figure 1), it is shown that the four different model showed different predictors to have the greatest influence. This suggests that the set of predictors with the greatest importance is highly sensitive to the model shown. An explanation is needed for why these predictors are actually the most important if their relative importance is reliant on the type of model being used.

MINOR COMMENTS

4. Typically, citations should appear within the sentence to which they refer. Throughout the text, they appear following the punctuation ending the sentence rather than before.

5. Lines 154-156: Not sure this sentence is needed. When describing the methods used in the study, it is unnecessary to describe others’ potential perceptions. If this is meant to be a genuine limitation of using the 70-30 split data, then it belongs in the discussion section.

6. Lines 291-292: Don’t the machine learning models also produce model coefficients? I recognize that the advantage is that these variable importance methods can prevent unimportant predictors from being included after model fitting, but perhaps this is the advantage rather than the fact that model parameter estimates are not present.

7. Table S2 shows logistic regression results, which show that six of the ten variables with “largest coefficients” have extraordinarily wide confidence intervals. This would typically be interpreted to mean that, within the context of this model, there is very little confidence in the direction and magnitude of association. Further, if these are model coefficients and not odds ratios, then the odds ratios for these six variables would show even wider confidence intervals. This may relate to why the four ML models selected different sets of most important predictors.

Some typos and grammatical errors exist. However, this is not distracting and only warrants routine editing. 

Author Response

Thank you for considering our paper " Table 2 Fallacy in Descriptive Epidemiology: Bringing Machine Learning to the Table" for publication into International Journal of Environmental Research and Public Health (IJERPH).  We appreciate the feedback provided by the reviewers. A revised manuscript is attached that addresses and carefully incorporates each of these comments. Below are our responses to the comments and feedback from the reviewers.

The authors present a method using Machine Learning to analyze descriptive epidemiologic data in a way that avoids the Table 2 fallacy, where all individual coefficients are erroneously interpreted as isolated, unconfounded causal effects without considering their context for inclusion in the model. Table 2 fallacy is unfortunately common in many contexts of data analysis, so I am glad to see research aiming to address this issue. I do think further clarity is needed regarding why it is the Machine Learning process that avoids Table 2 fallacy rather than the use of variable importance measures.

While I agree that naively running a “kitchen sink” regression model and treating every model coefficient as an interpretable isolated effect is not a best practice, more information and rationale is needed to convince the reader that Machine Learning is needed to avoid Table 2 fallacy as opposed to other methods of variable selection outside of Machine Learning.

Thank you for reviewing our paper.

MAJOR COMMENTS

  1. In order for the proposed Machine Learning process to be viewed as a viable alternative to typical multivariable regression models that is not subject to Table 2 fallacy, more explanation is needed regarding why the ML process is crucial. Other methods for assessing relative importance of predictors exist outside of machine learning. Currently, this paper provides good justification for why analyses beyond a multivariable regression are needed to compare relative impact of different predictors in order to identify ones that warrant further investigation. However, it is not clear why this ML method is needed and why variable importance cannot be assessed without it.

Thank you for this insightful comment. We agreed that variable importance can be assessed even in logistic regression models. However, we believe ML has some additional benefits that it allows for variable selection where p >> n (i.e. when there are more predictors than sample size), which a traditional logistic regression cannot accommodate. Furthermore, while it’s true variable importance can be calculated in regression models, it may also be harder to implement, and machine learning models such as trees and random forest can be naturally extended to calculate variable importance, making it more attractive. There are also less distributional assumptions from machine learning compared to the traditional regression models. Finally, as ML has proven effectiveness in predictive modeling, we believe for researchers who are already familiar with ML, these post-hoc procedures are straightforward to implement, and that they would complement an otherwise black-box predictive model by providing descriptive and interpretable findings  We believe these are all cases to make for advancing machine learning even though a traditional regression model could also work, these can be used in tandem. We have added a sentence that makes this very clear in Section 1.2 where traditional regression models can also obtain VI scores, but ML feels like a natural solution for the reasons given above.

  1. Section 1.2 mentioned multiple potential methods for assessing variable importance. The methods section does not specify what one was used or why it is a valid measure of influence of a predictor on the outcome of interest. Otherwise, this method is presented as a “black box” that is not subject to Table 2 fallacy because the authors claim that is the case.

Thank you for the clarification, all our variable importance scores are calculated using the model agnostic approach. We have made this more explicit and added an additional reference from Molnar (2022) to clarify this approach for interested readers.

  1. In the results (lines 211-224 and Figure 1), it is shown that the four different model showed different predictors to have the greatest influence. This suggests that the set of predictors with the greatest importance is highly sensitive to the model shown. An explanation is needed for why these predictors are actually the most important if their relative importance is reliant on the type of model being used.

Thank you, we agreed that this is a limitation. We have provided a framework in Table 1 that ideally will be able to reconcile the inconsistent results between different models. Specifically, we recommend the use of bootstrapping combined with nested cross-validation to quantify the uncertainty associated with the variable importance score. We always recommend the predictive performance as the best indicator of the model, although we also suggest making the conclusion based on the variables that are consistently selected by different models. We agreed these are some limitations that we need to be aware of, and currently there is no clear answer for which ML model is best suited for a particular question. However, we still think the proposed approach can be an improvement over traditional kitchen sink regression models.

MINOR COMMENTS

  1. Typically, citations should appear within the sentence to which they refer. Throughout the text, they appear following the punctuation ending the sentence rather than before.

Thank you this has been corrected.

  1. Lines 154-156: Not sure this sentence is needed. When describing the methods used in the study, it is unnecessary to describe others’ potential perceptions. If this is meant to be a genuine limitation of using the 70-30 split data, then it belongs in the discussion section.

Thank you, this has been removed and moved to the discussion.

  1. Lines 291-292: Don’t the machine learning models also produce model coefficients? I recognize that the advantage is that these variable importance methods can prevent unimportant predictors from being included after model fitting, but perhaps this is the advantage rather than the fact that model parameter estimates are not present.

Thank you for this comment. We did not mean to imply that ML does not produce model coefficients, however, we are suggesting that analysis from traditional regression models typically results in a saturated table with all model coefficients. ML would prevent this, especially we rarely received model coefficients directly.

  1. Table S2 shows logistic regression results, which show that six of the ten variables with “largest coefficients” have extraordinarily wide confidence intervals. This would typically be interpreted to mean that, within the context of this model, there is very little confidence in the direction and magnitude of association. Further, if these are model coefficients and not odds ratios, then the odds ratios for these six variables would show even wider confidence intervals. This may relate to why the four ML models selected different sets of most important predictors.

Thank you, this is another limitation we are facing, as we see with our bootstrapped CIs, we do receive very wide intervals, suggesting that we may produce inconsistent results. Given that the logistic regression also produces inconsistent results, this gives us more confidence in our results. Perhaps one can replicate an ML analysis with larger data with more robustness to examine if the results are more consistent across different bootstrap runs. This may help us determine if the inconsistency were due to the data or the ML process. Furthermore, we are conducting a descriptive epidemiology study, which are often aiming to provide preliminary findings that could inspire future, more in-depth & focused investigation. As such it is completely normal that we see these very wide CIs indicating uncertainty of individual importance and potentially complex interplay (interactions effects) amongst variables - these results call for further investigation with larger sample sizes. This point has been added into the discussion in section 4 of page 9.

Comments on the Quality of English Language

Some typos and grammatical errors exist. However, this is not distracting and only warrants routine editing. 

Reviewer 3 Report

It was a pleasure for me to read and review the manuscript titled "Table 2 Fallacy in Descriptive Epidemiology: Bringing Machine Learning to the Table" submitted for publication in "IJERPH" journal.

The article is original and well written. However, there are some changes that must be done to merit publication in this valuable journal. Here are some comments:

1. Please add more recommendations and/or conclusions in the end of your abstract

2. Line 29, please delate "(2022)"

3. Please be careful while using punctiation, specifically while using references, for example line 30 and 31.

4. I think you need at least to add a more general backround before starting to write about descriptive epidemiology

5. What is LASSO ? please write the full terms instead of the acronym the first time it appears in the text.

6. Line 137, same thing as my previous comment, Coronavirus Disease 2019 (COVID-19).

7. What was the package you used in R ?

8. Why is figure 3 included in both text and supplementary material ?

9. There are some cases, especially in low-middle income countries when applying a ML approach in not possible due to the quality of data provided by authorities, I think it's interesting to perform other kind of modeling,  Please refer to these papers and consider citing them:

https://doi.org/10.1038/s41598-022-09442-z

https://doi.org/10.3390/ijerph19159586

https://doi.org/10.1016/j.cmpb.2021.106301

https://doi.org/10.1155/2021/1250129

10. I think there is a lack of references, specifically in the discussion section, please try to compare your study with more references, I think you have to add about 5 additional references, not only those that I recommended but also others.

11. Normaly, you should respect the layout of the journal in your supplementary documents.

Apart from that, the proposed methodology and the entire document are interesting and pleasant to read. Again, I wanted to congratulate the authors.

I wish the authors good luck.

There are some minor corrections to be made. Nevertheless, the overall level of English is good.

Author Response

Thank you for considering our paper " Table 2 Fallacy in Descriptive Epidemiology: Bringing Machine Learning to the Table" for publication into International Journal of Environmental Research and Public Health (IJERPH).  We appreciate the feedback provided by the reviewers. A revised manuscript is attached that addresses and carefully incorporates each of these comments. Below are our responses to the comments and feedback from the reviewers.

Comments and Suggestions for Authors

It was a pleasure for me to read and review the manuscript titled "Table 2 Fallacy in Descriptive Epidemiology: Bringing Machine Learning to the Table" submitted for publication in "IJERPH" journal.

The article is original and well written. However, there are some changes that must be done to merit publication in this valuable journal. Here are some comments:

Thank you

  1. Please add more recommendations and/or conclusions in the end of your abstract

Thank you, we have added a sentence about inviting other researchers to use these methods.

  1. Line 29, please delate "(2022)"

Removed

  1. Please be careful while using punctiation, specifically while using references, for example line 30 and 31.

Thank you, this has been changed

  1. I think you need at least to add a more general backround before starting to write about descriptive epidemiology

Thank you for this comment. We have added a few sentences on the use of ML in public health, especially in the field of substance use on page 3 of section 1.2, which was also requested by another reviewer.

  1. What is LASSO ? please write the full terms instead of the acronym the first time it appears in the text.
  2. Line 137, same thing as my previous comment, Coronavirus Disease 2019 (COVID-19).

We have added the abbreviation for LASSO and COVID-19.

  1. What was the package you used in R ?

This was stated in line 199 to 202.

  1. Why is figure 3 included in both text and supplementary material ?

Thank you, to clarify these are two different Figures resulting from the interaction analysis. Figure 3 depicts the most interesting interaction between cocaine use x rurality, cannabis x suicidality on alcohol use, whereas the Supplemental in S3 shows the different most important interactions identified by the different ML models.

  1. There are some cases, especially in low-middle income countries when applying a ML approach in not possible due to the quality of data provided by authorities, I think it's interesting to perform other kind of modeling,  Please refer to these papers and consider citing them:

https://doi.org/10.1038/s41598-022-09442-z

https://doi.org/10.3390/ijerph19159586

https://doi.org/10.1016/j.cmpb.2021.106301

https://doi.org/10.1155/2021/1250129

  1. I think there is a lack of references, specifically in the discussion section, please try to compare your study with more references, I think you have to add about 5 additional references, not only those that I recommended but also others.

Thank you, more references have been added. We also added the discussion about the use of ML in settings with lower resources, and cited works on responsible use of AI and ML.

  1. Normally, you should respect the layout of the journal in your supplementary documents.

Apart from that, the proposed methodology and the entire document are interesting and pleasant to read. Again, I wanted to congratulate the authors.

I wish the authors good luck.

Thank you for reviewing our paper.

Comments on the Quality of English Language

There are some minor corrections to be made. Nevertheless, the overall level of English is good.

Thank you for this comment.

Round 2

Reviewer 1 Report

With the improvements that the authors have implemented, the work meets the established minimums. For me, the work must be accepted in its current form.

Author Response

Thank you for the comments. There were no further concerns raised by this reviewer.

Reviewer 2 Report

I appreciate the authors' consideration of my previous comments and for revising the manuscript. Overall, my comments have been adequately addressed. I only have small suggestions. 

My impression from reading this study is that the ability for these Machine Learning methods to perform variable selection and reduce the number of potential predictors is where much of the advantage to using them lies. If the authors agree that this is, at a high level, where the advantage comes from, it would be helpful to state this somewhere in the text rather than having the reader "read between the lines" to catch this. 

Figure 2 should be a single item with one caption. If there are concerns about Figures 2a, 2b, and 2c being too large as one item, then they can be split into three figures. 

I recommend one final proofreading for grammar and readability. Examples are the sentence on lines 66-68 (remove "one covariate" from line 67), line 157 ("Lists" or "A list"), and line 181 (missing period).

Overall readability is good. One close read for proofreading would be beneficial. 

Author Response

Comments and Suggestions for Authors

I appreciate the authors' consideration of my previous comments and for revising the manuscript. Overall, my comments have been adequately addressed. I only have small suggestions. 

My impression from reading this study is that the ability for these Machine Learning methods to perform variable selection and reduce the number of potential predictors is where much of the advantage to using them lies. If the authors agree that this is, at a high level, where the advantage comes from, it would be helpful to state this somewhere in the text rather than having the reader "read between the lines" to catch this. 

Thank you. We believe this is not the only advantage but we have made it clearer in the conclusion and discussion, on lines 310 and 408, we have discussed that the advantages of ML are automatic variable selection and variable importance scores.

Figure 2 should be a single item with one caption. If there are concerns about Figures 2a, 2b, and 2c being too large as one item, then they can be split into three figures. 

Thank you for this comment. We did split them as Figures 2a, 2b and 2c because they are too large as one item and each caption is explaining each respective figure, combining them into one caption would be too text for one caption. To avoid confusion, we have removed the overall Figure 2 caption, and referred to them as Figures 2a, 2b, and 2c. We believe this is the most appropriate as Figures 2a, 2b and 2c are all about PDPs from the different models, so it should go under one heading of Figure 2 with sub-headings of Figures 2a, 2b, and 2c, rather than presenting them separately as Figure 2, Figure 3 and Figure 4.

I recommend one final proofreading for grammar and readability. Examples are the sentence on lines 66-68 (remove "one covariate" from line 67), line 157 ("Lists" or "A list"), and line 181 (missing period).

Thank you, this has been fixed.